# COVID-19, Government Response, and Market Volatility: Evidence from the Asia-Pacific Developed and Developing Markets

**Izani Ibrahim [1], Kamilah Kamaludin [2],\* and Sheela Sundarasen [2]**

[1] Finance Department, College of Business Administration, Prince Sultan University, Rafha Street, Riyadh 11586, Saudi Arabia; profizani@psu.edu.sa

[2] Accounting Department, College of Business Administration, Prince Sultan University, Rafha Street, Riyadh 11586, Saudi Arabia; ssundarasen@psu.edu.sa

\* Correspondence: kkamaludin@psu.edu.sa

**Abstract:** This study examines the relationship between COVID-19, government response measures, and stock market volatilities for 11 developed and developing economies within the Asia-Pacific region. Our period of study is between 15 February–30 May 2020. Using the continuous wavelet transformation (CWT) analysis and plots and GJR-GARCH analysis, we examined the effects of the COVID-19 public health crisis and the corresponding government measures on the respective domestic equity markets volatilities. The CWT plots showed a varying level of market volatilities at different investment horizons. All the sample countries, except Japan, experienced very low or low volatility over the short-term horizons. In contrast, Vietnam, Malaysia, and Laos experienced medium volatility over the medium-term horizons. Finally, China, Japan, South Korea, Malaysia, and the Philippines experienced high volatility over the long-term horizons. The GJR-GARCH results further ascertain that market volatilities are affected by domestic events, notably, the COVID-19 government intervention measures. In most sample countries, the government measures significantly reduce market volatility in the domestic equity markets. Additionally, international events have also triggered market volatilities. Overall, our study offers several contributions and implications for practitioners and policymakers.

**Keywords:** COVID-19; Asia-Pacific; market volatility; government response; wavelet analysis

---

## 1. Introduction

The COVID-19 pandemic is pushing global economies into recession and reproducing financial market volatilities of the Great Depression of the 1930s and the Global Financial Crisis of 2008 (Sharif et al. 2020). The pandemic is presenting enormous challenges, causing many uncertainties and enacting sharp trade-offs. The crises are international but have intense local impacts as they are threatening people's livelihood and crippling markets and economies (Goodell 2020). Globally, governments have responded swiftly by sanctioning lockdowns and social isolation to contain the spread of the disease, especially after the pandemic declaration by the World Health Organization (WHO). In response to this global economic downturn, countries worldwide have also adopted wide-ranging and significantly large economic and financial stimulus packages to mitigate the destruction posed by the pandemic. Thus, the purpose of this study is to examine the relationship between COVID-19 daily infection cases, government response measures, and equity market volatilities for 11 developed and emerging economies within the Asia-Pacific region. Our period of study is between 15 February–30 May 2020. While earlier studies used statistical modeling, our analysis employs the wavelet spectrum approach to identify market volatility across time and frequency domains.

Limited understanding of the implications of the pandemic on the financial markets motivates many studies to gain insights into what caused the dramatic financial market movements at the height of the outbreak (Ashraf 2020; Baker et al. 2020; Gormsen and Koijen 2020; Zhang et al. 2020). One primary source of market volatility arises from the uncertainty and potential economic losses from the pandemic (Shanaev et al. 2020; Sharif et al. 2020; Zhang et al. 2020). Lockdowns and quarantines in many countries cause economic contraction and financial losses (Ali et al. 2020; Baker et al. 2020; Sharif et al. 2020). The demand for oil became severely affected by the reduction of logistics and transportation as many factories and suppliers halted operations, and air travel was suspended due to the closure of international borders and tourism (Mzoughi et al. 2020; Sharif et al. 2020). Baker et al. (2020) posited the effect of the interconnectedness and dependence of global economies and supply chains has severely impacted market volatility during this time.

The severity of the outbreak and its threat to human lives, whether the spread took place on international grounds (Albulescu 2020; Onali 2020) or individual countries (Albulescu 2020; Zhang et al. 2020), triggered market volatility (Baker et al. 2020). For example, Onali (2020) found evidence that the number of cases and deaths related to COVID-19 affected the volatility of the DowJones and S&P 500, even when it happened outside the US. Al-Awadhi et al. (2020) studied the Chinese stock market and presented evidence on the relations between the number of cases and deaths and the stock markets movement. Shanaev et al. (2020) found that expected infection peaks and local case growth affected stock market volatilities of the 51 equity markets that they studied. Ali et al. (2020) and Albulescu (2020) also found evidence that global financial markets volatilities increased once the casualty of COVID-19 started rising outside its epicenter, China.

The spread of the COVID-19 cases prompted government containment measures (Wagner 2020) in the form of restriction of movements and quarantines that halted production and all other economic activities. These containment measures are complemented with monetary and fiscal support to the society (Wagner 2020). Several studies have looked at the impact of government policies during the COVID-19 separately or jointly. For example, Zaremba et al. (2020) provide evidence that government containment measures (in the form of stringency policy) lead to volatility increase, which is separate from the pandemic itself. Restriction of movement prompts probable economic losses, which later will be mediated through economic intervention and relief. Such anticipation raises market volatility (Zaremba et al. 2020). On a similar note, Baker et al. (2020) also relate market volatility with investors' reactions to government containment policies. Wagner (2020) proposes that both government's containment measures and fiscal policy interventions reduce the COVID-19 shock (Wagner 2020).

Interestingly, while government response is anticipated to improve market sentiment (Sharif et al. 2020), Shanaev et al. (2020) found conversely. In their study of 51 equity markets, government response in the form of national lockdowns, monetary or fiscal stimulus appeared to be counterproductive, leading to economic losses and negative market returns. Instead, targeted regional lockdown is deemed more productive (Shanaev et al. 2020). Gormsen and Koijen (2020) documented a similar effect on the US market. Following the Black Monday of 2020, the US government had implemented a series of measures to appease the market. The economic relief program and fiscal stimulus packages, though improving stock returns, did not improve market expectations (Gormsen and Koijen 2020) as anticipated.

To this end, the situation created by COVID-19, both in the spread and severity of the outbreak and corresponding government measures, prompt economic uncertainty and potential financial losses. Investors' reaction to such a situation is evident through market volatility (Ali et al. 2020; Ashraf 2020; Baker et al. 2020; Mzoughi et al. 2020; Shanaev et al. 2020; Sharif et al. 2020). A growing number of studies are looking at the effects of the pandemic outbreak on major financial markets but overlooking the emerging markets, particularly in Asia. This paper shall examine the financial markets' reaction to the pandemic outbreak on domestic equity markets of several emerging and mature economies in Asia, including China. By using the wavelet power spectrum approach and the GJR-GARCH model, this paper examines and compares the relations among financial market

volatilities, COVID-19 daily infection cases, and the corresponding government responses. To the best of our knowledge, this paper is the first comparative study on stock volatilities that examines selected developed and developing markets of the Asia Pacific region during the COVID-19 pandemic using the wavelet methodology. Comparing the trends of these markets jointly is expected to bring clues as to how the health crisis affected the equity markets. Asian countries responded with measures to contain the epidemic as soon as news confirming its spread was released. Due to the proximity and close relations with China, Asian countries are at the forefront of COVID-19 exposure. True enough, all initial cases in Asia had been imported from Wuhan, Hubei, from Chinese tourists. While the global health crisis and containment are unfolding in each country, the financial markets have also seen unprecedented volatilities (Zhang et al. 2020) comparable to the 2008 Financial Crisis, prompted by several international or world events (Baker et al. 2020; Gormsen and Koijen 2020). Thus, this study's findings shall contribute to the understanding of the implication of health crises such as the COVID-19 pandemic and the corresponding government response to the equity market's volatility of the Asia-Pacific countries.

## 2. Materials and Methods

This paper analyzed COVID-19, government response to the COVID-19 pandemic, and equity market data from eight ASEAN countries; Indonesia, Laos, Malaysia, Myanmar, Philippines, Singapore, Thailand, and Vietnam. Additionally, China, Japan, and South Korea are also included in this study to enable comparison with more developed countries in the quantum of COVID-19 pandemic statistics and government responses. Brunei, though an ASEAN country, is excluded due to data unavailability.

Data used in this study are sourced from three databases, namely, the Worldometer website[1], the Oxford COVID-19 Government Response Tracker (OxCGRT) (Hale et al. 2020), and Bloomberg. Data span is from 15 February to 30 May 2020, where non-trading days are omitted. OxCGRT made available 18 indicators of the government's response to the COVID-19 pandemic. The government response index score is from 0 to 100. Data collected are from ordinal, numeric, and text indicators collected from over 150 countries. These data form the four main indices (Hale et al. 2020), namely, the overall government response index, stringency index, containment and health index, and economic support index. The indicators for these four indices are from four categories of indicators, namely, containment and closure (eight indicators), economic response (four indicators), health system (five indicators), and miscellaneous (one indicator). The response to most of the indicators are through an ordinal response (0,1,2,3), and, depending on the geographic scope of an indicator, an additional binary flag (using a 0 and 1) if the government's response is general (1) or targeted (0) to a specific location. This flag will create a "bonus" point in the calculation of the respective indices.

For each indicator, the sub-index score is calculated by the following,

$$I_{j,t} = 100 \frac{v_{j,t} - 0.5(F_j - f_{j,t})}{N_j} \tag{1}$$

where $I_{j,t}$ is the sub-index score for a given indicator $j$ at time $t$, $v_{j,t}$ is the recorded ordinal score, $F_j$ is a dichotomous score if the indicator has a flag variable (1) or not (0), $f_{j,t}$ is the values (0,1) of the flag variable, and $N_j$ is the maximum possible score of the ordinal indicator. From (1), it is clear that if an indicator has a flag variable and the geographical response is targeted, a penalty of 0.5 (out of the maximum score $N_j$) score is imposed. Note that the maximum value $I_{j,t}$ is 100. Finally, the score for an index $I_t$ at time $t$ is the average score of all indicators ($I_{j,t}$) included in the index score and is given by

---

[1] https://www.worldometers.info/coronavirus/.

$$I_t = \frac{1}{n} \sum_{j=1}^{n} I_{j,t} \qquad (2)$$

where $n$ is the total number of indicators in the index $I$.

### 2.1. Methodology

The multiscale nature of the time-series for financial and economic data requires different structures at different time scales (Khalfaoui and Boutahar 2011). This multiscale nature can be captured in time and frequency domains. Most studies on these time-series data focused only on the time scale, thus leaving out information content of the frequency domain, resulting in less informed investment decisions. Wavelets can adapt well to the unexpected and inexplicable change of the statistical analysis of economic and financial data (Ramsey 1999). Some of the advantages of using wavelet analysis include its ability to differentiate information on few data points, simplicity of application (Ramsey 1999), and flexibility in handling non-stationary data (Donoho et al. 1995).

#### 2.1.1. The Continuous Wavelet Transformation (CWT)

The CWT captures the time and frequency (or time signals) domains of the behavior of time-series data. This is done by looking at the spectral characteristics of the signals in the time domain. Wavelet or "daughter wavelet" $\psi_{\tau,s}(t)$, provides the wavelet coefficients for the "mother wavelet" $\psi(t)$, and is normalized defined as

$$\psi_{\tau,s}(t) = \frac{1}{\sqrt{|s|}} \psi\left(\frac{t-\tau}{s}\right) s \qquad (3)$$

with, $\tau \in \mathfrak{R}$, a location parameter and $s \neq 0$, a scale smoothing parameter.

For time-series $x(t)$, convoluting the function $\psi_{\tau,s}(t)$ with the series produces the transformation,

$$W_x(\tau,s) = \frac{1}{\sqrt{|s|}} \int_{-\infty}^{+\infty} x(t)\psi * \left(\frac{t-\tau}{s}\right) dt \qquad (4)$$

where * denotes the complex conjugate form. The mother wavelet $\psi(t)$ is used to generate other window functions at a location center $\tau$. As the window shifts through time, time information is obtained in the transformed domain. The scale, $s$, controls the length of the wavelet (referred to as daughter wavelet), giving the frequency information from the time-series data by dilating ($|s| > 1$) and compressing ($|s| < 1$) the series.

Wavelet power spectrum can be defined as $\left|W^x(\tau,s)\right|^2$ where the expectation $\left|W^x(\tau,s)\right|^2$ is equal $N \cdot |\hat{x}_k|^2$, $|\hat{x}_k|$ the discrete Fourier transformation (DFT) is defined as $\hat{x}_k = \frac{1}{N} \sum_{n=0}^{N-1} x_n e^{-2\pi i k n/N}$, where $k = 0$, ..., $N-1$ is the frequency index.

For a white-noise process, the expected value of $E(|\hat{x}_k|^2) = \frac{\sigma^2}{N}$, where $\sigma^2$ is the variance. Thus, $E\left(\left|W^x(\tau,s)\right|^2\right) = \sigma^2$ for all $n$ and $s$ (Torrence and Compo 1998).

This paper uses Morlet's wavelet, which is usually used for financial and economic data (Goupillaud et al. 1984). This is expressed as

$$\psi_\eta(t) = \pi^{-\frac{1}{4}} \left(e^{i\eta t} - e^{-\frac{\eta^2}{2}}\right) e^{-\frac{t^2}{2}} \qquad (5)$$

The function ensures the admissibility condition since it is negligible if $\eta \geq 5$. For $\eta \geq 5$, and since $e^{-\frac{\eta^2}{2}}$ is very small, we get Morlet's wavelet

$$\psi_\eta(t) = \pi^{-\frac{1}{4}} \left(e^{i\eta t}\right) e^{-\frac{t^2}{2}} \qquad (6)$$

### 2.1.2. The GJR-GARCH Model

Further analysis is made to study how the daily COVID-19 cases and the government response relate to the volatility of the stock market indexes in the ASEAN region and the developed market. To achieve this, we run the Glosten–Jagannathan–Runkle GJR-GARCH (1,1) model (Glosten et al. 1993) on the market indexes to obtain the conditional variance of the market indexes. The GJR-GARCH (1,1) is given by:

$$r_t = \mu + \varepsilon_t$$
$$\sigma_t^2 = \omega + (\alpha + \gamma I_{t-1})\varepsilon_{t-1}^2 + \beta\sigma_{t-1}^2$$
$$I_{t-1} = \begin{cases} 0 & if \ r_{t-1} \geq \mu \\ 1 & if \ r_{t-1} < \mu \end{cases} \tag{7}$$

It can be observed that GARCH, is a special case of GJR-GARCH, with $\gamma = 0$. The restriction of the parameters $\omega$, $\alpha$, $\gamma$, $\beta > 0$ is applied for the stability of the estimation. The GJR-GARCH (1,1) also captures the presence of volatility clustering and theorizes that the volatility at time $t$ is more likely to be higher if it is also high at time $t-1$. Another interesting property of this model is that the volatility itself is mean reverting towards $\sigma$ if,

$$\alpha + \frac{\gamma}{2} + \beta < 1. \tag{8}$$

The parameter $\gamma$ can be interpreted as the leverage effect, specifically the effect of positive and negative volatility shocks, where the positive shock has a higher impact on the conditional variance. The results of the GJR-GARCH (1,1) are given in Table 2 below.

The conditional variance $\sigma_t^2$ is extracted from the GJR-GARCH (1,1) model and used as a dependent variable in the following linear regression;

$$\sigma_t^2 = \alpha_0 + \alpha_1(daily\ Covid\ cases_t) + \alpha_2(government\ response_t) + \nu_t \tag{9}$$

Finally, the Newey and West (1987) method is used to control for heteroskedasticity and autocorrelation problems through the heteroskedasticity and autocorrelation consistent (HAC) estimations.

## 3. Results

### 3.1. Continuous Wavelet Transform Results

Figure 1 shows COVID-19 cases in the respective countries, the government response index, and the power spectrum (variance) of the stock market index. The top part of each graph examines the volatility of the stock market index using a continuous wavelet transform (CWT) method. The bottom part of the graph combines the time series of COVID-19 daily new cases and government response index for 11 countries, both developed; Japan and Singapore, and developing countries; South Korea, Malaysia, China, Indonesia, Philippines, Thailand, Laos, Vietnam, and Myanmar.

The *x*-axis of Figure 1 graphs (top and bottom parts) shows the period selected for this study, which coincides with the spread of the COVID-19 cases, 15 February–30 May 2020. The *y*-axis for the blue line time-series graph represents the number of COVID-19 daily cases, and the *y*-axis of the red line time-series graph represents the government response index score. The *x*-axis of the wavelet plot is consistent with the other two time-series graphs (15 February–30 May 2020). The color code on the right of the wavelet plot represents the volatility spectrum, ranging from blue (low volatility) to yellow (high volatility). The *y*-axis on the left of the wavelet plot represents the level of frequency or time-periods, covering short-term or high-frequency bands (approximately 0 to 4 days), medium-term, or medium frequency bands (approximately 5 to 8 days), and long-term horizons or low-frequency bands (9 days onwards).

The CWT plots describe the equity market volatility in both time and frequency. The regions inside the blue contour plotted in darker color represent low volatility, and as it becomes lighter and

warmer, it represents higher volatility or greater risk. In particular, the islands marked by a bold line in a light shade indicate the significance of influence according to the value of the power spectrum (volatility) described by the color codes, during the time, concentrated at specific frequency regions.

The top part of the CWT plot for all countries indicates very low volatility, denoted by the dark blue shading throughout the sample period. The medium and long-term horizons indicate relatively low, medium, and high volatility throughout the sample period. This is inferred from the blue, green, and yellow shades on the contour on the bottom half of the continuous wavelet transformation plots.

The CWT plots suggest that over the short-term horizons, Vietnam and Indonesia experienced very low volatility, whereas Singapore, South Korea, China, Philippines, Laos, Myanmar, Thailand, and Malaysia experienced low volatility. Next, Vietnam, Malaysia, and Laos experienced medium-low and medium volatility over the medium-term horizons. Lastly, China, Japan, South Korea, Malaysia, and the Philippines experienced high volatility over the long-term horizons, as evident from the yellow island hanging over the contour.

Table 1 below further interprets the CWT plot of the equity market indexes, daily COVID-19 cases, and government response to the COVID-19 pandemic in Figure 1. In addition, details about market volatility, the range of standard deviation, the period of significant volatility, frequency bands, and horizons, and the underlying reasons for market volatilities are also included. Appendix A also provides detailed explanations of the CWT plot, COVID-19 graph, and government response index.

### 3.2. GJR-GARCH (1,1)

This section provides the results of our analysis using a standard time-series modeling approach examining the relations between the daily COVID-19 cases and the government response and the volatility of the stock market indexes in the ASEAN region and the developed market.

Table 2 below provides the estimates of the GJR-GARCH (1,1) model in which the conditional variance is extracted and further regressed using the Newey-West method.

Table 3 below provides the results of the Newey and West (1987) regression analysis. This method is selected as it can control for heteroskedasticity and autocorrelation problems through the heteroskedasticity and autocorrelation consistent (HAC) estimations.

At the 10% significance level, our findings indicate that government responses are affecting market volatility in all domestic indexes, except for the South Korean equity market. In particular, government responses are found to have negative relations with market volatility, suggesting that government responses reduce market volatility for all equity markets except for China. In China, the government response appears to significantly induce greater market volatility.

In terms of the COVID-19 infection cases, the results are rather heterogeneous on the effect of the COVID-19 cases on market volatility. At the 5% significance level, the COVID-19 infection cases are increasing market volatility in China and Thailand. On the contrary, in Japan, Laos, and the Philippines the COVID-19 infection cases are reducing market volatility. For the remaining countries, at the 10% level, there are no significant relations between COVID-19 infection cases with market volatility.

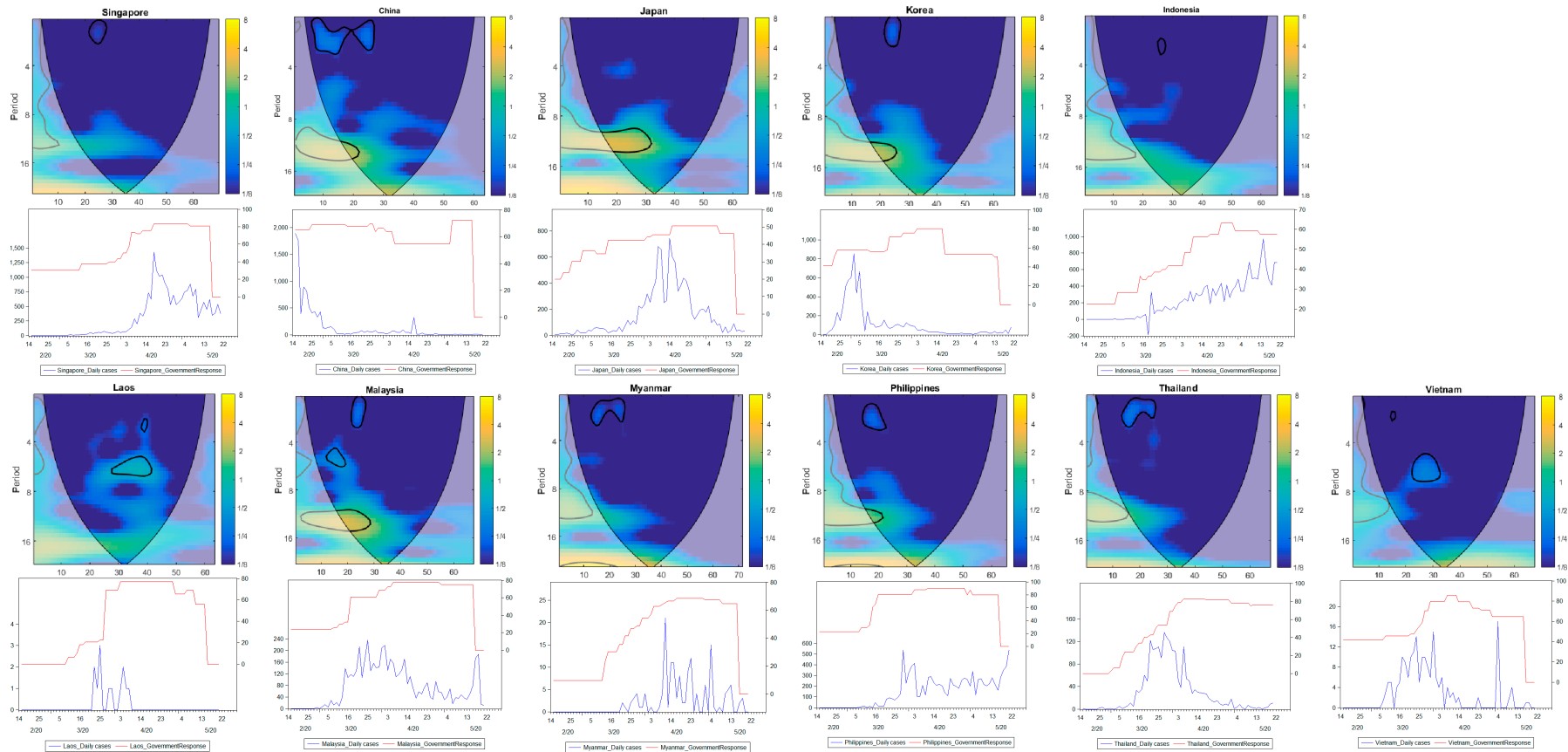

**Figure 1.** Continuous wavelet transform (CWT) of equity market indexes, daily COVID-19 cases, and government response to COVID-19 pandemic.

**Table 1.** Detail analysis of significant market volatilities of the CWT plots.

| Country | Significant Volatility | Range of SD | Period of Significant Volatility | Freq. Bands | Horizon | Govt. Response | COVID-19 Trend | Factors or Reasons for Volatilities |
|---|---|---|---|---|---|---|---|---|
| Vietnam | very low volatility | ⅛ | 6–10 March | 1–2 | short-term | An increasing index score of 42 to 46 | 2nd stage—a resurgence of new cases | Second waves detected and moderate government response—Overlapping with international events |
| Indonesia | very low volatility | ⅛ | 20–26 March | 2–3 | short-term | An increasing index score of 37 to 40 | Early-stage—increasing trend | The spread of COVID-19 cases and moderate government response |
| Singapore | low volatility | ⅛–¼ | 17–26 March | 0–2 | short-term | Index score of 38 | Early-stage—increasing trend | The spread of COVID-19 cases and moderate government response |
| South Korea | low volatility | ¼–½ | 18–26 March | 0–2 | short-term | An increasing index score of 56 to 72 | Mid-stage—decreasing trend | Reduction of COVID-19 cases and high government response |
| China | low volatility | ¼–½ | 24 February–24 March | 0–3 | short-term | Index score at 69 | End-stage—decreasing trend | Reduction of COVID-19 cases and high government response |
| Philippines | low volatility | ¼–½ | 6–23 March | 1–3 | short-term | An increasing index score of 22 to 80 | Very early stage—an increasing trend | Initial reaction to COVID-19 cases and rapid government response |
| Laos | low volatility | ¼–½ | 20–22 April | 2–3 | short-term | An index score of 77 | End-stage—zero cases | Containment of COVID-19 infection through a high government response |
| Myanmar | low volatility | ¼–½ | 4–23 March | 0–2 | short-term | An increasing index score of 10 to 35 | Very early stage—First 2 cases detected on 23 March | Anticipation period before COVID-19 spread in the country—Overlapping with international events |
| Thailand | low volatility | ¼–½ | 5–25 March | 0–3 | short-term | An increasing index score of 6 to 50 | Early-stage—increasing trend | The spread of COVID-19 cases and high government response—Overlapping with international events |
| Malaysia | low volatility | ¼–½ | 18–25 March | 0–3 | short-term | An increasing index score of 32 to 61 | Early-stage—increasing trend | COVID-19 spread (local transmission) and high government response |
| Vietnam | medium-low volatility | ½–1 | 18 March–1 April | 5–7 | medium-term | An increasing index score of 46 to 80 | Third stage—increasing trend | High government response at peaked of COVID-19 cases |

**Table 1.** *Cont.*

| Country | Significant Volatility | Range of SD | Period of Significant Volatility | Freq. Bands | Horizon | Govt. Response | COVID-19 Trend | Factors or Reasons for Volatilities |
|---|---|---|---|---|---|---|---|---|
| Malaysia | medium volatility | 1–2 | 4–16 March | 4–6 | medium-term | An increasing index score of 24 to 32 | Early-stage—increasing trend | Political turmoil |
| Laos | medium volatility | 1–2 | 27 March–23 April | 5–7 | medium-term | An increasing index score of 22 to 77 | Early period—mixed trend | High government response at initial reaction to a spike of COVID-19 cases |
| China | high volatility | 2–4 | 9–18 March | 11–14 | long-term | An index score of 69 | End-stage—declining trend | Timeline is overlapping with international events |
| Japan | high volatility | 2–4 | 9 March–3 April | 9–13 | long-term | An increasing index score of 35 to 44 | Early-stage—increasing trend | Timeline is overlapping with international events |
| South Korea | high volatility | 2–4 | 9–25 March | 12–15 | long-term | An increasing index score of 58 to 72 | Mid-stage—declining trend | Timeline is overlapping with international events |
| Malaysia | high volatility | 2–4 | 9–27 March | 11–15 | long-term | An increasing index score of 32 to 61 | Early-stage—increasing trend | Timeline is overlapping with international events |
| Philippines | high volatility | 2–4 | 9–20 March | 10–13 | long-term | An increasing index score of 22 to 80 | Very early stage—an increasing trend | The timeline is overlapping with international events. |

**Table 2.** Results for the GJR-GARCH (1,1) with country market indexes as the dependent variable.

| Country | Equation | Independent Variables | Coefficient | Std. Error | t-Value | t-Prob |
|---|---|---|---|---|---|---|
| China | Mean | Cst(M) | 2835.486 ** | 5.908 | 479.900 | 0.000 |
| | Variance | Cst(V) | 157.100 * | 84.032 | 1.870 | 0.066 |
| | | ARCH(Alpha1) | 0.769 ** | 0.173 | 4.435 | 0.000 |
| | | GARCH(Beta1) | 0.226 | 0.142 | 1.596 | 0.115 |
| | | GJR(Gamma1) | −0.019 | 0.185 | −0.103 | 0.918 |
| Indonesia | Mean | Cst(M) | 4590.870 ** | 16.978 | 270.400 | 0.000 |
| | Variance | Cst(V) | 869.320 ** | 407.290 | 2.134 | 0.037 |
| | | ARCH(Alpha1) | 0.309 ** | 0.116 | 2.670 | 0.010 |
| | | GARCH(Beta1) | 0.574 ** | 0.145 | 3.955 | 0.000 |
| | | GJR(Gamma1) | 0.038 | 0.196 | 0.193 | 0.848 |
| Japan | Mean | Cst(M) | 19.801 ** | 0.331 | 59.830 | 0.000 |
| | Variance | Cst(V) | 0.109 | 0.088 | 1.234 | 0.222 |
| | | ARCH(Alpha1) | 0.823 ** | 0.147 | 5.593 | 0.000 |
| | | GARCH(Beta1) | 0.063 | 0.053 | 1.188 | 0.239 |
| | | GJR(Gamma1) | 0.271 | 0.367 | 0.739 | 0.463 |
| Korea | Mean | Cst(M) | 1930.973 ** | 5.351 | 360.900 | 0.000 |
| | Variance | Cst(V) | 498.527 | 442.600 | 1.126 | 0.265 |
| | | ARCH(Alpha1) | 0.900 ** | 0.153 | 5.876 | 0.000 |
| | | GARCH(Beta1) | −0.004 | 0.047 | −0.084 | 0.933 |
| | | GJR(Gamma1) | 0.267 | 0.269 | 0.991 | 0.326 |
| Laos | Mean | Cst(M) | 620.694 ** | 0.563 | 1103.000 | 0.000 |
| | Variance | Cst(V) | 2.899 | 2.150 | 1.349 | 0.182 |
| | | ARCH(Alpha1) | 0.706 ** | 0.098 | 7.188 | 0.000 |
| | | GARCH(Beta1) | 0.159 ** | 0.062 | 2.573 | 0.013 |
| | | GJR(Gamma1) | 1.192 * | 0.637 | 1.872 | 0.066 |
| Malaysia | Mean | Cst(M) | 1386.135 ** | 4.651 | 298.100 | 0.000 |
| | Variance | Cst(V) | 117.720 ** | 37.055 | 3.177 | 0.002 |
| | | ARCH(Alpha1) | 0.988 ** | 0.171 | 5.774 | 0.000 |
| | | GARCH(Beta1) | 0.017 | 0.102 | 0.165 | 0.870 |
| | | GJR(Gamma1) | 0.097 | 0.300 | 0.325 | 0.747 |
| Myanmar | Mean | Cst(M) | 111.980 ** | 0.342 | 327.200 | 0.000 |
| | Variance | Cst(V) | 0.261 | 0.224 | 1.166 | 0.248 |
| | | ARCH(Alpha1) | 0.543 ** | 0.129 | 4.223 | 0.000 |
| | | GARCH(Beta1) | 0.467 ** | 0.126 | 3.710 | 0.000 |
| | | GJR(Gamma1) | −0.060 | 0.162 | −0.368 | 0.714 |
| Philippines | Mean | Cst(M) | 5583.420 ** | 23.226 | 240.400 | 0.000 |
| | Variance | Cst(V) | 223.271 | 439.170 | 0.508 | 0.613 |
| | | ARCH(Alpha1) | 0.173 * | 0.089 | 1.951 | 0.056 |
| | | GARCH(Beta1) | 0.785 ** | 0.101 | 7.800 | 0.000 |
| | | GJR(Gamma1) | −0.146 ** | 0.068 | −2.158 | 0.035 |
| Singapore | Mean | Cst(M) | 2562.246 ** | 15.944 | 160.700 | 0.000 |
| | Variance | Cst(V) | 422.448 | 482.860 | 0.875 | 0.385 |
| | | ARCH(Alpha1) | 0.363 | 0.245 | 1.481 | 0.143 |
| | | GARCH(Beta1) | 0.462 | 0.350 | 1.319 | 0.192 |
| | | GJR(Gamma1) | 0.143 | 0.265 | 0.540 | 0.591 |
| Thailand | Mean | Cst(M) | 857.434 ** | 24.235 | 35.380 | 0.000 |
| | Variance | Cst(V) | 124.572 | 152.740 | 0.816 | 0.418 |
| | | ARCH(Alpha1) | 0.865 | 0.578 | 1.498 | 0.139 |
| | | GARCH(Beta1) | 0.032 | 0.526 | 0.060 | 0.952 |
| | | GJR(Gamma1) | 0.434 | 0.933 | 0.465 | 0.643 |
| Vietnam | Mean | Cst(M) | 770.208 ** | 1.805 | 426.700 | 0.000 |
| | Variance | Cst(V) | 27.578 * | 14.700 | 1.876 | 0.065 |
| | | ARCH(Alpha1) | 1.097 ** | 0.110 | 9.976 | 0.000 |
| | | GARCH(Beta1) | 0.008 | 0.014 | 0.593 | 0.556 |
| | | GJR(Gamma1) | 0.150 | 0.283 | 0.529 | 0.599 |

Significance level: ** $p < 0.05$, * $p < 0.10$.

**Table 3.** Regression of $\sigma_t^2 = \alpha_0 + \alpha_1 (daily\ Covid\ cases_t) + \alpha_2 (government\ response_t) + v_t$.

| Country | Independent Variables | Coef. | Std. Err. | t-Stat | $p > t$ | [95% Conf. Interval] | |
|---|---|---|---|---|---|---|---|
| China | Daily COVID-19 cases | 0.101 ** | 0.049 | 2.05 | 0.044 | 0.003 | 0.200 |
| | Government Response | 1.644 ** | 0.452 | 3.64 | 0.001 | 0.743 | 2.546 |
| | Constant | −16.630 | 23.690 | −0.70 | 0.485 | −63.903 | 30.643 |
| Indonesia | Daily COVID-19 cases | 0.000 | 0.000 | 0.63 | 0.529 | 0.000 | 0.001 |
| | Government Response | −0.022 ** | 0.005 | −3.94 | 0.000 | −0.033 | −0.011 |
| | Constant | 1.218 ** | 0.221 | 5.50 | 0.000 | 0.776 | 1.660 |
| Japan | Daily COVID-19 cases | −0.005 ** | 0.001 | −4.16 | 0.000 | −0.007 | −0.003 |
| | Government Response | −0.076 * | 0.042 | −1.79 | 0.078 | −0.160 | 0.009 |
| | Constant | 6.481 ** | 1.859 | 3.49 | 0.001 | 2.767 | 10.194 |
| Korea | Daily COVID-19 cases | −0.143 | 0.144 | −1.00 | 0.323 | −0.430 | 0.144 |
| | Government Response | 3.990 | 4.571 | 0.87 | 0.386 | −5.147 | 13.128 |
| | Constant | 129.916 | 278.452 | 0.47 | 0.642 | −426.703 | 686.534 |
| Laos | Daily COVID-19 cases | −3.346 ** | 1.342 | −2.49 | 0.015 | −6.027 | −0.665 |
| | Government Response | −0.194 ** | 0.050 | −3.84 | 0.000 | −0.295 | −0.093 |
| | Constant | 18.944 ** | 3.119 | 6.07 | 0.000 | 12.713 | 25.175 |
| Malaysia | Daily COVID-19 cases | 0.035 | 0.081 | 0.43 | 0.666 | −0.126 | 0.196 |
| | Government Response | −1.360 ** | 0.277 | −4.91 | 0.000 | −1.914 | −0.807 |
| | Constant | 129.644 ** | 20.442 | 6.34 | 0.000 | 88.830 | 170.458 |
| Philippines | Daily COVID-19 cases | −0.210 ** | 0.058 | −3.61 | 0.001 | −0.326 | −0.094 |
| | Government Response | −1.194 ** | 0.303 | −3.95 | 0.000 | −1.799 | −0.589 |
| | Constant | 163.757 ** | 28.521 | 5.74 | 0.000 | 106.779 | 220.735 |
| Singapore | Daily COVID-19 cases | −0.006 | 0.005 | −1.40 | 0.166 | −0.015 | 0.003 |
| | Government Response | −0.124 * | 0.069 | −1.80 | 0.076 | −0.261 | 0.013 |
| | Constant | 15.680 ** | 3.549 | 4.42 | 0.000 | 8.596 | 22.763 |
| Thailand | Daily COVID-19 cases | 1.580 ** | 0.391 | 4.04 | 0.000 | 0.800 | 2.360 |
| | Government Response | −1.691 ** | 0.318 | −5.32 | 0.000 | −2.326 | −1.057 |
| | Constant | 143.196 ** | 24.283 | 5.90 | 0.000 | 94.728 | 191.664 |
| Vietnam | Daily COVID-19 cases | −1.842 | 1.312 | −1.40 | 0.165 | −4.459 | 0.776 |
| | Government Response | −1.554 ** | 0.421 | −3.69 | 0.000 | −2.394 | −0.714 |
| | Constant | 166.550 ** | 30.085 | 5.54 | 0.000 | 106.516 | 226.584 |
| Myanmar | Daily COVID-19 cases | 0.187 | 0.288 | 0.65 | 0.518 | −0.387 | 0.761 |
| | Government Response | −0.491 ** | 0.095 | −5.18 | 0.000 | −0.680 | −0.302 |
| | Constant | 145.770 ** | 5.842 | 24.95 | 0.000 | 134.122 | 157.419 |

significance level: ** $p < 0.05$, * $p < 0.10$.

## 4. Discussion

In this section, we discussed several factors that influence equity market volatilities of the sample countries. Based on the empirical evidence, visual inspection and corroborated by the GJR-GARCH model, the results indicate that market volatilities are affected by the domestic events as they influence perceived risks and uncertainty from the COVID-19 situation and government measures. Additionally, we also indicate that international events have also triggered market volatilities.

The country-specific or domestic events involving the COVID-19 situation have caused varying levels of market volatility for the sample countries. This is evident from the visual inspection of the CWT plots in Figure 1 and further supported by the regression results in Table 3. The market movements over the short and/or medium-term horizons result from domestic events reflecting either the increase or decrease of the COVID-19 daily cases and/or government response in containing the COVID-19 spread in each country.

For instance, the Malaysian equity market exhibits low volatility following the increased government response in combating the sudden surge of COVID-19 cases on the Malaysian ground. On 16 March, the Malaysian government announced that the Movement Control Order (MCO;

lockdown) would be imposed from 18 March onwards. The equity market volatility is attributed to the reaction to the MCO and its impact on the country's economy. The regression estimate in Table 3 indicates a significant negative relationship suggesting that government measures reduce Malaysian market volatility.[2]

The Malaysian equity market may have also experienced other domestic events that influence market volatility. The CWT plot indicates an occurrence of medium volatility over the medium-term horizons which coincides with the political turmoil and uncertainties over the new government's future amid the worsening of the COVID-19 outbreak in Malaysia.[3]

The Singaporean equity market reacts to an intermediate level of government measures following the COVID-19 outbreak in Singapore. Market participants seemed to have factored in the impact of the COVID-19 outbreak on the country's economy. On 26 March, Singapore's Ministry of Trade and Industry had declared an estimated economic contraction by 2.2% (Ministry of Trade and Industry 2020) measured year on year, in the first quarter which was their worst since the 2008 global financial crisis[4]. Despite the unfavorable economic outlook, the regression estimate in Table 3 suggests that the government measures have significantly reduced market volatility for the Singaporean index.

Figure 1 indicates that Myanmar's COVID-19 cases are almost non-existent until 23 March when the first case is identified. The government response has steadily increased to its maximum level during this period. The regression estimate in Table 3 indicates that the government measures have significantly reduced market volatility for Myanmar's equity index.

The Philippines' equity market reacts to the COVID-19 outbreak and government measures with low volatility. This is supported by significant regression results in Table 3 that indicate government responses and COVID-19 infection cases reduce market volatility. COVID-19 infection cases are still in their early stage in the Philippines, and the rapid government measure may mitigate the fear from the COVID-19 outbreak which in turn significantly reduces market volatility for the Philippines market.

The government response in Laos was initiated well before the first COVID-19 case is detected. Laos equity market reacts to Laos' government response approaching its maximum level. When total cases were still less than 10, the government increased its response rate to 69 on 30 March and 77 on 6 April. There are no new cases in Laos after 20 April. Like the Philippines, rapid government response coupled with modest COVID-19 cases reduce Laos market volatility in Table 3.

The Vietnamese equity market reacts with low volatility following a resurgence of new COVID-19 cases. On 6 March, an infected patient was identified returning from Europe. This case triggers the second wave and the Second Stage of government response[5]. Vietnam had been free from any cases for the past 10 days before this. Significant medium volatility on the Vietnamese equity market coincides with the peaking of COVID-19 cases with total cases surpassing 100 and the government's declaration of their Third Stage response in fighting the coronavirus. While the regression estimate does not support any evidence of the COVID-19 infection cases inducing market volatility but the government measures on the Vietnamese ground appear to significantly reduce market volatility.

Interestingly, the CWT plots in Figure 1 also indicate that the Vietnamese and Laos equity markets experience a medium level of volatility over their medium-term horizons. Arguably, smaller developing economies experience relatively higher volatility and risks than larger economies, which is in line with earlier findings (Aggarwal et al. 1999; Bekaert and Urias 1999). During a pandemic, small economies are in crisis mode. Their main priorities are to contain the spread of the virus, assess and augment their healthcare capacity to prepare for an unexpected upsurge of infection bouts, and support their vulnerable population economically. For instance, Vietnam had responded in an extreme manner

---

2     https://www.thestar.com.my/business/business-news/2020/03/18/volatile-market.
3     https://www.thestar.com.my/business/business-news/2020/03/03/double-whammy.
4     https://www.channelnewsasia.com/news/business/singapore-gdp-growth-slows-advance-estimate-covid-19-12577146.
5     https://ourworldindata.org/covid-exemplar-vietnam.

knowing that their medical system may be overwhelmed[6] by even a mild spread of the virus. Besides, their many years of experience being a neighbor and on receiving end of other viruses from China had taught them the serious implication of the outbreak. Due to that, the fear of the pandemic could trigger higher volatility to Vietnam and Laos more than any other equity markets.

The CWT plot in Figure 1 indicates that the Thailand equity market experiences a low level of volatility following an increasing trend of government response and the beginning phase of the COVID-19 infection outbreak. The regression results in Table 3 further suggest that the steady increase of government response reduces market volatility, whereas the increasing trend of the COVID-19 cases induces market volatility.

The CWT plot also shows that the Indonesian equity market reacts with low volatility following a moderate government response. Similar to other countries in this study, the significant regression results suggest that the government response reduces market volatility for the Indonesian market.

The Table 3 regression estimate suggests that the imposition of government intervention measures reduces market volatility in domestic equity markets for all countries except for China. While the CWT plot indicates that the Chinese equity market's low volatility corresponds with the declining number of COVID-19 cases and the easing prospects from the strict government measures, the regression results appear to suggest otherwise. The regression results indicate that both government measures and COVID-19 infection cases significantly induce market volatility. While the spread and fear from the COVID-19 infection may have lingered and affected market volatility, the government measures in China also lead to market volatility. This observation seems contradictory with other countries in Asia. Arguably, the strict and draconian measures imposed by the Chinese government may have not been well received by the general public causing a heightened level of anxiety and fear and subsequently high market volatility.

The CWT plot indicates that the Japanese equity market experiences significantly high volatility over its long-term horizons. The regression results in Table 3, however, indicate that its domestic events in form of the government response towards COVID-19 and the daily infection cases are reducing market volatility. Though Japan is close to China in proximity, unlike China, the COVID-19 cases do not seem to spike uncontrollably in Japan. During the period of volatility, average cases in Japan are still under control and because of that the government measures and the infection cases reduce market volatility.

Lastly, the CWT plot indicates that the South Korean equity market shows significant low volatility during the period that corresponds with incremental government response in the form of economic stimulus to businesses and households[7] coupled with a declining number of daily new cases. On 24 March, the South Korean government announced a financial stabilization plan amounting to KRW100 trillion or 5.3% of its GDP[8]. However, the regression estimates in Table 3 do not find any significant relations between market volatility with the respective domestic events on the South Korean market.

Overall, our empirical evidence suggests that market volatilities are influenced by domestic events. Our results that relate the COVID-19 daily infection cases with market volatilities for several of our sample countries are in line with findings by Onali (2020), Al-Awadhi et al. (2020), and Shanaev et al. (2020).

Further, our evidence also strongly indicates that in most of our sample countries, the corresponding government intervention measures undertaken by respective countries reduce market volatility. We argue that market participants gain confidence from government measures, whether in the form of stringency, containment, or economic stimulus, in mitigating the pandemic's negative effects. While our findings are in line with the proposition by Wagner (2020), which supports the positive implication of

---

[6] https://www.bbc.com/news/world-asia-52628283.
[7] https://en.yna.co.kr/view/AEN20200324008251320.
[8] https://www.imf.org/en/Topics/imf-and-covid19/Policy-Responses-to-COVID-19#R.

government intervention measures to the financial markets, it is mostly in contradiction with other studies, such as Zaremba et al. (2020), Ashraf (2020) and Shanaev et al. (2020) who generally find that government's intervention measures pose an adverse implication on the financial markets.

However, the Chinese equity market that is found to react adversely to the government measures is in line with Zaremba et al. (2020), Ashraf (2020), and Shanaev et al. (2020)'s findings. According to Nurunnabi et al. (2020), the Chinese government executed severe control measures, which led to a 6.8% economic contraction in the first quarter of 2020. The stringency measure in China was very high, which, between January to 31 May, on several occasions, had peaked beyond level 80 (100 is the strictest response possible) (Nurunnabi et al. 2020). Chinese authoritarian, aggressive, and extreme government measures were proven to be successful but critics argued it happened at the expense of the citizens' human rights (Kupferschmidt and Cohen 2020). Massive lockdowns and electronic surveillance are notably coercive in nature, yet public compliance was still in order despite the extreme measures (Kupferschmidt and Cohen 2020). Arguably, the political system in China was different from the other countries and the regression results that suggest government measures and COVID-19 cases are inducing market volatility are justified in this given context. This finding on the Chinese equity market is supported by Zaremba et al. (2020) who also found that government intervention measures (in the form of stringency policy) lead to volatility increase.

Our finding also complements earlier studies that have examined the effect of the COVID-19 situation and market volatility (Ali et al. 2020; Baker et al. 2020; Mzoughi et al. 2020; Shanaev et al. 2020; Sharif et al. 2020). Domestic events reflecting either the increase or decrease of the COVID-19 daily cases and/or government response towards containing the COVID-19 spread affected the domestic equity markets inducing low volatility over the short-term horizons.

While our regression analysis has been helpful in ascertaining the effects of domestic events on market volatility and explains some of the observations of the CWT plot. The CWT plot also indicates several instances of significant medium and high volatility which we argue may not directly be attributed to domestic events. As such, we infer our findings to the occurrence of international events that overlap with the period of significant volatility in the CWT plots.

Two significant international events have caused a severe impact on the financial markets worldwide; (1) oil price disputes and (2) declaration of the pandemic by the World Health Organization (Ali et al. 2020; Ashraf 2020; Sharif et al. 2020; Zhang et al. 2020). The COVID-19 pandemic has affected oil demand and prices since the beginning of the year. Due to a break-up in the dialogue between OPEC and Russia to agree on oil production cuts, on 8 March, Saudi Arabia announced unexpected price discounts, which triggered a free-fall in oil prices, the most significant drop since 1991. On 9 March, global stock markets experienced a dramatic fall due to the oil price dispute and the panic and fear over the outspread of COVID-19 worldwide, which the World Health Organization officially declared as the global pandemic on 11 March. This period is known as the March 2020 Stock Market Crash[9], paralleling the 2008 Financial Crisis (Mazur et al. 2020; Shanaev et al. 2020).

These international events have affected the market volatility of our sample countries. Based on the timing of the events (the week beginning from 9 March), our CWT plots in Figure 1 indicate that China, South Korea, Japan, Malaysia, and the Philippines exhibit significantly high volatility over their long-term horizons. Additionally, the international events may have also affected Thailand, Myanmar, and Vietnam, as indicated by their significant low volatility overlapping with the 9 March market crash. For example, in Thailand, it was reported that on 9 March, Thailand's equity market contracted by 8% with investors cashing out approximately THB78.36 billion[10].

Interestingly, the Singaporean equity market does not seem to be affected by high volatility during this period. Unlike other developed economies observed in this study, Singapore's equity market is

---

[9]　March 2020 Stock Market Crash also records single-day extreme events: Black Monday I (9 March), Black Thursday (12 March) and Black Monday II (16 March).

[10]　https://www.bangkokpost.com/business/1875059/set-tumbles-ptt-group-shares-lead-losses.

buffered by the external shocks. Arguably, this finding suggests several implications. First, it suggests that Singapore is an independent market whereby shocks transmitted from events that transpired from inside the country affected its volatility more than from outside. This observation is in line with earlier studies suggesting that markets are more susceptible to domestic volatility shocks than international events (Aggarwal et al. 1999; Bekaert and Harvey 1997; Gérard et al. 2003; Yarovaya et al. 2020). A plausible explanation for this could be due to the type of market participants of Singaporean equity markets. Arguably, the literature suggests that developed markets attract informed traders (rational) based on fundamental factors rather than behavioral factors such as sentiment (Thampanya et al. 2020). The market reaction prompted by fear of the pandemic is arguably a behavioral sentiment and irrational (Kit 2020). Instead, we find evidence to suggest that fundamental factors influenced by domestic events drive Singapore's market volatility. Specifically, the government intervention measures and responses to the COVID-19 pandemic are indicators of macroeconomic fundamentals informing rational investors' decisions (Shanaev et al. 2020). This particular observation is in line with Thampanya et al. (2020), who find evidence that fundamental factors play critical roles in influencing equity market volatility in Singapore.

## 5. Conclusions and Implication of the Study

This study uses the wavelet approach and GJR-GARCH analysis to examine the impact of COVID-19 and government measures on the equity market volatilities of selected developing and developed countries from the Asia-Pacific region. The period of study is from 15 February to 30 May 2020. Countries included in this study are Indonesia, Laos, Malaysia, Myanmar, the Philippines, Singapore, Thailand, Vietnam, China, Japan, and South Korea.

Our study provides several contributions and implications to the literature. First, the equity market volatilities are examined across several emerging and developed markets using a novel approach, the wavelet analysis. The wavelet analysis provides insights into two critical dimensions, that is, time and frequency. Using the wavelet analysis and plots, we examine the effects of the public health crisis and the corresponding government measures on its respective domestic equity markets. As the unfolding of the national health crisis overlaps with some critical international events, such as the Market Crash 2020 resulting from the oil dispute and the pandemic declaration, we observe the effects of these events on the domestic equity markets.

Second, this study further substantiates the evidence that the public health crisis and government measures significantly influence market volatilities in our sample countries. Conversely, not all countries were affected by international events. Thus, downplaying the effects of the national health crisis over the financial contagion or spillover arguments may potentially misguide market participants' asset allocation strategies. Similarly, Asian markets that appear less susceptible to external shocks, present an opportunity for international portfolio diversification investment strategy in this region (Aggarwal et al. 1999; Yarovaya et al. 2020).

Lastly, it has become apparent that government response to specific health crises influences the level of risk or uncertainty to the investors. While some studies indicate that higher government response may be associated with greater risks and uncertainty (Zaremba et al. 2020), our findings indicate mixed results. For most countries in this study, the government intervention measures are perceived positively, hence, reducing market volatilities. However, extreme and strict government measures may also be counterproductive and lead to increased market volatility. Thus, this finding presents an important consideration for policymakers and practitioners in future health crises and emergencies. Saving lives are and safeguarding livelihood are both critically imperative. Careful trade-off needs to be considered on a timely basis when confronted with public health emergencies.

**Author Contributions:** Conceptualization, I.I.; Data curation, K.K.; Formal analysis, I.I.; Methodology, I.I.; Software, I.I.; Writing—original draft, K.K.; Writing—review & editing, S.S. All authors have read and agreed to the published version of the manuscript.

**Funding:** This research received no external funding.

**Acknowledgments:** The author would like to thank Prince Sultan University for their support.

**Conflicts of Interest:** The authors declare no conflict of interest.

## Appendix A

This section provides a more in-depth discussion of each country based on the CWT plot, COVID-19 graph, and government response index.

### Appendix A.1. China

Chinese equity markets are generally volatile over the medium and long-term horizons, as indicated by the various color shades on the contour (light blue to light yellow). Two islands emerged on the contour between 24 February–24 March, exhibiting significantly low volatility over a high-frequency band of 0–3 days and between 9–18 March significantly high volatility over a low-frequency band of 11–14 days.

Cases in China were peaking on 12 February with daily new cases of 14,000 a day. At the beginning of our sample period, daily new cases were showing a declining trend. On 21 February, daily new cases were less than 1000 a day. The government response had been initiated very early on, and by 25 February had already approached its maximum score of 69. As the cases continued to decline, on 8 April, the Chinese government had reduced its government response to below 60 from the stringency aspect. This continued for about a month, and on 11 May, the government response score was increased to 72. The incremental government response was also from the stringency aspect. While the COVID-19 cases continued to decline, the Chinese equity market appears to be significantly volatile, displaying a low level of risks over the short-term horizon.

The short-term, low market volatility corresponded with a government response with an index score of 69 and a declining number of daily new cases from 508 to 47, with an average of 140 cases a day. The long-term, high market volatility corresponded with a government response of 69 and a declining number of daily new cases of 19 to 34, with an average of 20 cases a day. The period of high market volatility also coincided with significant international events—oil disputes and WHO pandemic declaration.

### Appendix A.2. Japan

Japan's equity market indicates moderate to high volatility throughout March and April over a medium and long-term horizon but very low volatility over the short-term horizon. Between 9 March to 3 April, significantly high volatility is observed over the long-term horizons of 9–13 day frequency bands.

The daily new COVID-19 cases have been on the rise in the first half of the sample period, and a sudden increase was observed in the week beginning on 9 April. The highest number of cases exceeding 600 was reported on 15–16 April. From mid-May onwards, daily new cases have declined to less than 100. The government response in Japan has been steadily on the rise from an index score of 20 (18 February), 30 (25 February), 40 (16 March), and 50 (16 April). The maximum government response was observed following the sudden increase of daily new cases on 16 April. On 16 April, the Japanese PM declared a nationwide state of emergency and increased economic support for broader emerging markets and developing countries through the Poverty, Reduction, and Growth Trust (PRGT). The government lifted the state of emergency partially in mid-May and entirely at the end of May.

The long-term, high market volatility corresponded with a government response index score of 35 to 44 and an increasing number of daily new cases from 28 to 318 with an average of 95 cases a day. Government measure appears moderate compared to other countries, and the equity market is relatively stable and low risks except between 9 March to 3 April. Similar to China, the period of high volatility seemed to coincide with significant international events—oil disputes and WHO pandemic declaration

*Appendix A.3. South Korea*

South Korean equity market displays moderate and high risks beginning from the last week of February throughout March over the long-term horizons but mostly low risks over the short and medium-term horizons. Between 18–16 March, significant low volatility is observed over the short-term horizons of 0–2 days frequency bands, and between 9–25 March, significantly high volatility over the long-term horizons of 12–15 days frequency bands.

At the beginning of the outbreak, the rapid government response was observed from an index score of 40 (18 February) to 60 (24 February) in less than a week. From 23 March, the government response was increased to 70 and 80 (6 April), particularly from the stringency and containment aspects, before reducing it back to around 50 (20 April). Daily COVID-19 cases have been on the rise from February to the first week of March. The maximum number of cases, exceeding 800 (3 March) overlapped with a government response index of 60. As government response continues to rise, the number of daily cases continues to fall. By 16 March, daily cases were under 100. The government response seemed to be effective in flattening the curve.

The short-term, low market volatility corresponded with a government response with an index score of 56 to 72 and a declining number of new cases of approximately 97 a day. The long-term, high market volatility corresponded with a government response from an index score of 58 to 72 and a declining number of daily new cases from 165 to 100, with an average of 107 cases. Like China and Japan, the period of high market volatility coincided with significant international events –oil disputes and the WHO pandemic declaration.

*Appendix A.4. Indonesia*

The Indonesian equity market has low volatility over the short-term, low to medium volatility over the medium-term, and high volatility over the long-term horizons. Between 20–26 March, significant low volatility was detected over high-frequency bands of 2–3 days.

COVID-19 cases were identified somewhat later (6 March) in Indonesia compared to other neighboring countries. The average daily new cases in March are 55, in April it is 300, and in May it is 500.

Daily new cases were showing an increasing trend overlapping with an increasing government response. The maximum response is reported at an index score of 63 for around a week at the end of April and stabilizes around 57 for the rest of the sample period.

The short-term, very low market volatility corresponded with a government response with an index score of 37 to 40 and a sudden surge of daily new cases from 60 to 103 and continues to rise to date.

*Appendix A.5. Laos*

Laos equity market has low volatility over the short term and medium volatility over the medium-term, and high volatility over the long-term horizons from 5 March to 5 May. Between 27 March–23 April, a significant medium volatility is detected in the mid-sample period over 5–7 days frequency bands. Between 20–22 April, significant low volatility is detected over high-frequency bands of 2–3 days.

Government response with an index score of 22 was initiated well before the first case was detected in Laos. The first case was detected on 24 March. When total cases were still less than 10, the government increased its response to 69 (30 March) and a few days later at 77 (6 April). Around mid-May, the government response was reduced to 56 for a week before removing all measures on 22 May. By then, there were no new cases since 20 April. Government measure is safely removed after continuous zero cases for about a month.

The medium-term, medium market volatility corresponded with a government response with an index score of 22 to 77 and an increasing number of total daily new cases from 6 to 19. The short-term, low market volatility corresponded with a government response of 77 and zero number of daily new

cases from 20 April onwards. The significant volatility overlapped with Laos' government approaching its maximum response to contain COVID-19 outspread. Moreover, the low volatility corresponded with zero cases reported for Laos, beginning from 20 April onwards, indicating the government's success to stop the virus from spreading in the country.

### *Appendix A.6. Malaysia*

The Malaysian equity market has low volatility over the short-term horizons and medium to high volatility over the medium and long-term horizons from the end of February throughout March and April. During this period, significant equity market volatility is detected over the short, medium, and long-term horizons. Between 18–25 March, significantly low volatility was identified over 0–3 day frequency bands. Two weeks earlier, between 4–16 March, significant medium volatility was evident over 4–6 days frequency bands, and between 9–27 March, significantly high volatility over 11–15 day frequency bands.

The short-term, low market volatility corresponded with an increase of government response from an index score of 32 to 61 and an increasing number of daily cases from 117 to 172, with an average of 142 cases a day.

The medium-term, medium market volatility corresponded with a government response from an index score of 24 to 32 and an increasing number of daily cases from 14 to 138, with an average of 31 cases a day.

Volatility on the short-term horizons seemed to overlap with a sudden increase in daily cases on 16 March. During the same time, the government response was increased from an index score of 32 to 61 with its MCO announcement. The significant medium volatility seemed to correspond with Malaysia's political turmoil (sudden change of ruling party).

The long-term, high market volatility corresponded with a government response from an index score of 26 to 61 and an increasing number of daily cases from 18 to 130, with an average of 104 cases a day. Like China, Japan, and South Korea, the period of high market volatility coincided with significant international events—oil disputes and the WHO pandemic declaration.

### *Appendix A.7. Myanmar*

Myanmar's equity market has low volatility throughout the sample period over the short and medium-term horizons and medium to high volatility over the long-term horizons between 3 March–23 April. Between 4–23 March, significant low volatility is observed over 0–2 day frequency bands.

Myanmar's COVID-19 cases were almost non-existent until 23 March. When the first cases were identified, the government response steadily increased from an index score of 10 (18 February), 35 (20 March), 55 (1 April) to 69 (17 April). Myanmar's cases were relatively very low, with a total of 200 cases throughout the sample period. Government measure was removed by 26 May as the number of daily cases reported were less than five for the past three weeks.

The short-term, low market volatility corresponded with an increase of government response from an index score of 10 to 35 and an increasing number of daily cases from 0 to 2. The first case was detected on 23 March.

The significant volatility before the spread of COVID-19 in Myanmar may result from the anticipation of the effect of the cases and its implication to the country's modest economy, which may have induced investors' perceived risks and volatility. Another plausible explanation is that the significant low volatility emerging on the short-term horizon coincides with the timeline of significant international events, such as the oil disputes and WHO pandemic declaration.

### *Appendix A.8. Singapore*

The Singaporean equity market has low volatility throughout the sample period over the short and medium-term horizons and low to medium volatility over the long-term horizons between

7 March–3 April. Between 17–26 March, significant low volatility was detected over 0–2 day frequency bands.

The number of COVID-19 cases for Singapore has been relatively under control, and the government response has an index score of 30 at the beginning of the sample period throughout 12 March. When WHO declares it as a pandemic, the Singaporean government responded accordingly with a steady increase of the index score from 40 (27 March), 50 (3 April), 74 (8 April) to 83 (20 April), and remains at this level for several weeks. On 12 May, the government reduced its response to 81 and completed the removal of all measures two weeks later (26 May). The number of cases had also shown very little increase in the beginning but a sharp increase during the second week of April (7 April) with cases in hundreds and thousands by the third week of April. Corresponding government measures seemed to be in place on a timely basis.

The short-term, low market volatility corresponded with a government response with an index score of 38 and an increasing number of daily cases from 23 to 52, with an average of 46 cases. The sharp increase in COVID-19 cases did not affect the market volatility as investors were confident about the government response, and community transmission was still under control.

*Appendix A.9. Thailand*

Thailand's equity market has very low volatility over the short and medium-term horizons and low, medium, and high volatility over the long-term horizons from 4 March to 14 April. Between 5–25 March, significantly low volatility over 0–3 day frequency bands was identified.

The government response was seen as a steady increase from an index score of 6 (5 March), 24 (10 March), 30 (16 March), and 43 (23 March). This corresponds with a relatively low number of incremental COVID-19 cases in Thailand, whereby during this period, maximum cases were only 60 from containment and stringency aspects. However, when cases had increased to over 100 on 23 March onwards, the government responded sharply from an index score of 50 (25 March) to 83 (8 April) within two weeks to contain the virus spread. On 1 April, the government response included an economic support component. By 9 April, cases were under control (50 and less), and a steady decline was seen. From 27 April till the end of the sample period, daily cases were less than 10, but government response remained high at 76.

The short-term, low market volatility corresponded with government response with an index score of 6 to 50 and an increasing number of daily cases from 4 to 106, with average cases of 38. Though cases were still relatively low during the period of volatility, the increasing government response signals the government's readiness in facing the tumultuous period over the coming weeks during the height of the virus spread. The period of market volatility also coincided with significant international events—oil disputes and WHO pandemic declaration.

*Appendix A.10. Philippines*

Philippines equity market has low volatility over the short and medium-term horizons and medium to high volatility over the long-term horizons between 1 March–23 April. Between 6–23 March, significant low volatility over 1–3 day frequency bands was observed, and between 9–20 March, significantly high volatility over the long-term horizons or 10–13 day frequency bands was detected.

Philippines COVID-19 cases peaked in the week beginning on 30 March, and the highest number of cases were reported on 31 March, with daily cases exceeding 500. The government response has been steady at a score of 22 until 16 March it increased to 61 and 19 March to 80 and 6 April to 88 and 13 April to 90. Government response at 90 is by far the highest compared to other countries under investigation. When the sharp increase of COVID-19 cases emerged at the end of March, the government response was already very high, approaching its maximum. Once maximum measures were imposed at 90, cases were on a decline with an average of approximately 200 cases a day. However, when the government reduced its response rate to 80 and completely removed all measures on 26 May, cases seemed to be on the rise again rather than leveling off or declining (perhaps the second waves).

The short-term, low market volatility corresponded with an increase in government response from an index score of 22 to 80 and an increasing number of daily cases ranging from 22 to 82 with an average of 17. Cases were rapidly increasing at the end of March onwards. Market volatility may be an initial reaction to COVID-19 cases and the steady increase of government response in containing the virus (from 22 to 80) during this period.

The long-term, high market volatility corresponded with an increase of government response from an index score of 22 to 80 and an increasing number of daily cases with an average of 11 and also the international events unfolding during that time.

*Appendix A.11. Vietnam*

The Vietnam equity market has low volatility over the short and medium-term horizons and low, medium, and high volatility over the medium and long-term horizons. Between 6–10 March, significant low volatility is detected over 1–2 day frequency bands and low-to-medium volatility between 18 March–1 April over 5–7 day frequency bands.

The government response had an index score of 42 at the beginning of the sample period when total cases were less than 16. Once an infected patient was detected on 6 March, breaking the three weeks of zero daily cases, the government response was increased to an index score of 46 (6 March), 59 (27 March), 71 (30 March), and 80 (1 April) from the stringency and containment aspects. Vietnam's maximum government response was at 86, and at this point, total cases were around 266. From April 15 onwards, the government response was reduced as zero cases were reported throughout the remaining period (though few exceptions, most of the time, daily cases were zero). On 26 May, all government measure was lifted entirely. Vietnam reported total cases of 327 until the end of the study period.

The short-term, low market volatility corresponded with government response from an index score of 42 to 46 and an increasing number of daily cases ranging from 1 to 10.

The medium-term, medium-low market volatility corresponded with an increase of government response from an index score of 46 to 80 and an increasing number of total cases ranging from 76 to 218. On 23 March, Vietnam's cases broke 100 in total. This may have induced uncertainty and volatility in Vietnam's equity market. Though cases were still relatively low, the volatility may reflect the peaking of COVID-19 cases in Vietnam. The significant volatility on the medium-term overlapped with government response with an index score of 80. The period of low volatility could also coincide with significant international events such as the oil dispute.

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
