# Peer review of "COVID-19, Government Response, and Market Volatility: Evidence from the Asia-Pacific Developed and Developing Markets"

_economies, doi:10.3390/economies8040105_

Round 1

Reviewer 1 Report

Summary: The paper “COVID-19, Government Response and Market Volatility: Evidence from the Asia-Pacific developed and developing markets” studies stock market volatility and government responses in emerging markets during the first Covid19 wave. Most countries, except Japan, experienced fairly low levels of volatility even at different investment horizons.

General appraisal: The research question is potentially interesting, i.e. “What are the effects of governments’ Covid19 responses on market volatility”. But the paper underdelivers on the results. Not only are no statistical results derived between government response and market volatility, the paper also lacks depth in the sense that the graphical results are not investigated further in a rigorous empirical way. E.g. what is the difference between Japan and the other countries?, Are the drivers suggested in section 4 really the ones driving the results?,… .

Main comments:

Ideally, the paper would connect the four dimensions of analysis in the following way. How do government interventions (red line in Fig. 1) impact a) on cases (blue line in Fig.1), b) on volatility (top graphs in Fig. 1) and c) on the different frequencies (top graphs in Fig. 1). The methodology used might not be able to derive causality but even correlations might be very interesting. Also, is it clear that the effect goes from government intervention to volatility or is it a mix of government intervention and the number of cases which affect volatility? The “in-depth discussion” in the Appendix goes into this direction but would need to be substantiated with an empirical analysis.

Section 4 is a mix of potential explanations (domestic events, Perceived uncertainty and financial market integration). While it is a good thing that the authors thought about many potential reasons for the results they find, none of the explanations is tested empirically. Section 4 is therefore at best an informed discussion but without any impact for a policy makers and academic literature. To make the claims the paper sets out to make, these potential explanations would need to be tested empirically.

Minor comments:

  1. Why this peculiar methodology? Why not a classic case study? What are the gains? Given the fairly non standard approach, lines 133-137 are not enough to justify the usage of the methodology.
  2. The positioning of the paper in the literature needs to be clearer. What is different from Onali (2020), Gormsen and Koijen (2020), …? Is this the first paper looking at stock volatility in emerging markets?
  3. How do the results of this paper compare to results of other papers (e.g. developed vs. emerging markets)?
  4. Lines 71, 79, improve citation.
  5. The paper jumps between present and past tense. Unify.
  6. I would abstain from statements like “experienced significant very low volatility” as no confidence intervals are defined and tested.
  7. I am not sure which policy the journal follows but the authors names are shown on page 14.

Author Response

Reviewer 1

Summary: The paper “COVID-19, Government Response and Market Volatility: Evidence from the Asia-Pacific developed and developing markets” studies stock market volatility and government responses in emerging markets during the first Covid19 wave. Most countries, except Japan, experienced fairly low levels of volatility even at different investment horizons.

Noted.

General appraisal: The research question is potentially interesting, i.e. “What are the effects of governments’ Covid19 responses on market volatility”. But the paper underdelivers on the results. Not only are no statistical results derived between government response and market volatility, the paper also lacks depth in the sense that the graphical results are not investigated further in a rigorous empirical way. E.g. what is the difference between Japan and the other countries? Are the drivers suggested in section 4 really the ones driving the results?

Thank you for the comment. To enhance the wavelet methodologies, we adopted GJR-GARCH model to further reinforce our earlier findings and analyses of the CWT plots. We empirically tested the relations of market volatilities, daily COVID-19 infection cases and the government responses using the GJR-GARCH model.

See revised section 2.1.2, 3.2, 4.

Main comments:

Ideally, the paper would connect the four dimensions of analysis in the following way. How do government interventions (red line in Fig. 1) impact a) on cases (blue line in Fig.1), b) on volatility (top graphs in Fig. 1) and c) on the different frequencies (top graphs in Fig. 1).

The methodology used might not be able to derive causality but even correlations might be very interesting. Also, is it clear that the effect goes from government intervention to volatility or is it a mix of government intervention and the number of cases which affect volatility? The “in-depth discussion” in the Appendix goes into this direction but would need to be substantiated with an empirical analysis.

 Section 4 is a mix of potential explanations (domestic events, Perceived uncertainty and financial market integration). While it is a good thing that the authors thought about many potential reasons for the results they find, none of the explanations is tested empirically. Section 4 is therefore at best an informed discussion but without any impact for a policy makers and academic literature. To make the claims the paper sets out to make, these potential explanations would need to be tested empirically.

Thank you for your comment. The purpose of the study is not to look at the impact of the government interventions on COVID-19 cases but instead the relationship among market volatilities, COVID-19 cases and government interventions.

The initial study was not intended to look at causality. Nevertheless, we have now extended our analysis as suggested. We adopted the GJR-GARCH model and regression of the conditional variance to explore the relations among market volatilities, government responses and COVID-19 daily infection cases. Thus, now this study is substantiated with empirical analysis.

We have now revised Section 4 by incorporating only relevant sections as part of the discussion. This should further streamline the scope of the paper.

Minor comments:

Why this peculiar methodology? Why not a classic case study? What are the gains? Given the fairly non standard approach, lines 133-137 are not enough to justify the usage of the methodology.

The positioning of the paper in the literature needs to be clearer. What is different from Onali (2020), Gormsen and Koijen (2020), …? Is this the first paper looking at stock volatility in emerging markets?

How do the results of this paper compare to results of other papers (e.g. developed vs. emerging markets)?

Lines 71, 79, improve citation.

The paper jumps between present and past tense. Unify.

I would abstain from statements like “experienced significant very low volatility” as no confidence intervals are defined and tested.

I am not sure which policy the journal follows but the authors names are shown on page 14.

We have further elaborated the justification for using the wavelet approach. In addition, we have also combined the wavelet analysis with econometric analysis which is notable in forecasting market volatilities the GARCH model.

Please see lines 131-139, highlighted in yellow.

To the best of our knowledge, this paper is the first comparative study on stock volatilities that examines selected developed and developing markets of the Asia Pacific region during the COVID-19 pandemic using the wavelet methodology. Please see lines 91-93, highlighted in yellow

Please see discussion in Section 4.

Done

Done

Done

Removed

Reviewer 2 Report

The topic of the paper is very interesting, and the motivation provided by the authors is appealing. Nevertheless, to my view the manuscript lacks the standards to be published in a peer-reviewed journal:

  1. The authors seem to be keen to apply a given methodology, without providing any argumentation why this methodology is suitable for the question at hand, and how does it compare to alternatives in the literature.
  2. The visual presentation is fine as a descriptive way of showing the some intuitions in the estimations. But it is not appropriate for scientific testing of hypothesis: other variables can be determining the dynamics of all the selected variables in the plots. In order to test the link between volatility, stringency and incidence of the pandemic, a formal model, with control variables, is needed. Otherwise, nothing can be learnt from the exercise.
  3. In this regard, section 4 is purely speculative.

is too speculative. 

Author Response

Reviewer 2

 The topic of the paper is very interesting, and the motivation provided by the authors is appealing. Nevertheless, to my view the manuscript lacks the standards to be published in a peer-reviewed journal:

Noted.

The authors seem to be keen to apply a given methodology, without providing any argumentation why this methodology is suitable for the question at hand, and how does it compare to alternatives in the literature.

Thank you. We have further elaborated the justification for using the wavelet approach. In addition, we have also combined the wavelet analysis with econometric analysis which is notable in forecasting market volatilities the GARCH model.

Please see lines 131-139, highlighted in yellow.

The visual presentation is fine as a descriptive way of showing the some intuitions in the estimations. But it is not appropriate for scientific testing of hypothesis: other variables can be determining the dynamics of all the selected variables in the plots. In order to test the link between volatility, stringency and incidence of the pandemic, a formal model, with control variables, is needed. Otherwise, nothing can be learnt from the exercise.

This study does not intend to test any formal hypothesis. We have now extended our analysis as suggested. We adopted the GJR-GARCH model and regression of the conditional variance to explore the relations among market volatilities, government responses and COVID-19 daily infection cases. Thus, now this study is substantiated with empirical analysis.

See revised section 2.1.2, 3.2, 4.

In this regard, section 4 is purely speculative.

With the GJR-GARCH analysis, this study is now empirically tested and not speculative.

Section 4 is revised.

Reviewer 3 Report

Paper’s title : COVID-19, Government Response and Market Volatility: Evidence from the Asia-Pacific developed and developing markets

The present research assesses the linkage between COVID-19, government response measures, and stock market volatilities for different developed and developing countries within the Asia-Pacific region.

First I want to congratulate the authors for this paper. I found it really interesting, comprehensive and original. Moreover, you implement original techniques that allow to capture the hidden features in the relationship in question and this seems important in times of unprecedented uncertainty and rising volatility. Even though the terms uncertainty, volatility and risk have usually been used interchangeably, it seems a very good idea to distinguish between the three since the volatility measures do not consider the forecastable components of variations, and thus, they cannot completely reflect uncertainty. What I understand from the title and the text is that authors want to analyze the  interconnectedness of stock market volatility. However, there is no indication about the volatility measure used. What they used are the stock price indices, not their volatilities. Regarding the issue of the dynamic relationships between different Asian stock markets in this emergency situation, such investigation seems of utmost importance. Despite the quality of the paper, some points deserve further explanations. In particular, the paper lacks clarity on some aspects, which I will discuss in more details below.

Specific comments

  1. The authors should state more clearly the research question in the introduction.
  2. The authors should justify better the conducted methodology and the choice of the time span examined ; The period can be extended to September 2020 while considering the increasing risk of second wave pandemic.
  3. For  the volatility measure, I think that authors used the standard deviation. If so, the authors’ contribution is well seen. In other words, one does not see easily what the work of others is and what is the incremental contribution that your work brings in measuring volatility (risk) in times of heightened uncertainty.
  4. I believe that the sub-section section dealing with the methodology (in particular, the continuous wavelet) needs to be redesigned in order to facilitate the reader’s comprehension. The authors should develop this section because it is crucial for understanding.
  5. I think also that the usefulness and the relevance of analyzing the relationship among market volatilities using graphical tools is not well developed.
  6. Results and conclusions are sound but authors did not compare their results with previous work in order to clarify the contribution authors made.

Final assessment and recommendation

In the end, I believe that the paper deserves attention and that the empirical analysis is well conducted. Therefore, I suggest that the paper is revised according to the comments listed above and resubmitted for further consideration.

L

Author Response

Reviewer 3

The present research assesses the linkage between COVID-19, government response measures, and stock market volatilities for different developed and developing countries within the Asia-Pacific region.

Noted

First, I want to congratulate the authors for this paper. I found it really interesting, comprehensive and original. Moreover, you implement original techniques that allow to capture the hidden features in the relationship in question and this seems important in times of unprecedented uncertainty and rising volatility. Even though the terms uncertainty, volatility and risk have usually been used interchangeably, it seems a very good idea to distinguish between the three since the volatility measures do not consider the forecastable components of variations, and thus, they cannot completely reflect uncertainty.

What I understand from the title and the text is that authors want to analyze the interconnectedness of stock market volatility.

However, there is no indication about the volatility measure used. What they used are the stock price indices, not their volatilities.

Regarding the issue of the dynamic relationships between different Asian stock markets in this emergency situation, such investigation seems of utmost importance. Despite the quality of the paper, some points deserve further explanations. In particular, the paper lacks clarity on some aspects, which I will discuss in more details below.

Thank you.

Thank you for your comment. The purpose of the study is not to look at the interconnectedness of stock market volatility.  It examines the relationship among market volatilities, COVID-19 cases and government intervention in selected developed and developing equity markets of the Asia-Pacific region during the COVID-19 pandemic.

The measure of volatility is provided by the wavelet power spectrum which is the variance of the stock price indices. With reference to the wavelet graph, Torrence and Compo (1998) also documented this as the variance (volatility).

Thank you.

Specific comments

 The authors should state more clearly the research question in the introduction.

The authors should justify better the conducted methodology and the choice of the time span examined ;

The period can be extended to September 2020 while considering the increasing risk of second wave pandemic.

For  the volatility measure, I think that authors used the standard deviation. If so, the authors’ contribution is well seen. In other words, one does not see easily what the work of others is and what is the incremental contribution that your work brings in measuring volatility (risk) in times of heightened uncertainty.

I believe that the sub-section section dealing with the methodology (in particular, the continuous wavelet) needs to be redesigned in order to facilitate the reader’s comprehension. The authors should develop this section because it is crucial for understanding.

I think also that the usefulness and the relevance of analyzing the relationship among market volatilities using graphical tools is not well developed.

Results and conclusions are sound but authors did not compare their results with previous work in order to clarify the contribution authors made.

The research question has been addressed in highlighted section, lines 31-37  in the introduction.

Please refer to highlighted section, lines 34-37

We believed that the second and subsequent waves will have a different impact on the equity market volatilities and government responses. Thank you for highlighting this. It has already been considered for our next paper.

The measure of volatility is provided by the wavelet power spectrum which is the variance of the stock price indices. With reference to the wavelet graph, Torrence and Compo (1998) also documented this as the variance (volatility).

Thank you for the comment. Section 2 has been updated.

We have extended our graphical analysis with GJR-GARCH method to enhance the usefulness and relevance of the findings.

Refer to the updated section 4.

Final assessment and recommendation

In the end, I believe that the paper deserves attention and that the empirical analysis is well conducted. Therefore, I suggest that the paper is revised according to the comments listed above and resubmitted for further consideration.

Thank you.

Round 2

Reviewer 1 Report

While I am not able to evaluate the appropriateness of the methodology (GARCH) used to derive the additional results in section 3.2, I appreciate the additional analysis.

Main comments:

  1. What is the contribution of the paper? In the abstract it says "This study examines the relationship between COVID-19, government response measures, and stock market volatilities [...]." The paper goes on to show that following an economic shock, Covid19, volatility differs slightly depending on country and maturity. This is rather unsurprising. That is why I tried in my first referee report to push the authors to show what are the drivers of these results. This would be a lot more interesting.
  2. The overall message is somewhat confusing. At the end of the abstract the authors state "Overall, our study offers several contributions and implications for practitioners and policymakers." but at the same time they state in their replies "The purpose of the study is not to look at the impact of the government interventions on COVID-19 cases [...]". If it is not the purpose of the study to look at causality between Covid related government interventions on volatility than they cannot claim in the abstract that they derive policy implications.
  3. The revised section 4 is better now. The analysis derives correlations between government interventions and volatility. This section goes into the direction of explaining the why of differences between countries and maturities. Clearly there is no causality but potentially the Covid19 shock and the staggered responses of governments might lend itself to a difference in differences analysis which would go some way into the direction of causality. The authors should think harder to push section 4 and make this the main part of the paper. This would help addressing my comments 1 and 2.

Minor comments:

  • No page numbering.
  • There are almost 7 pages of continuous tables. It does not make for great readability.
